# Debating the Future of Sickle Cell Disease Curative Therapy: Haploidentical Hematopoietic Stem Cell Transplantation vs. Gene Therapy

**DOI:** 10.3390/jcm11164775

**Published:** 2022-08-16

**Authors:** Adetola A. Kassim, Alexis Leonard

**Affiliations:** 1Department of Medicine, Division of Hematology/Oncology, Vanderbilt Meharry Sickle Cell Center of Excellence, Vanderbilt University School of Medicine, Nashville, TN 37232, USA; 2Cellular and Molecular Therapeutics Branch, National Heart, Lung, and Blood Institute, National Institutes of Health, Bethesda, MD 20810, USA; 3Division of Hematology, Children’s National Hospital, Washington, DC 20010, USA

**Keywords:** allogeneic transplantation, autologous transplantation, CRISPR/Cas9, gene therapy, haploidentical, hematopoietic stem cell transplantation, sickle cell disease

## Abstract

Hematopoietic stem cell transplantation (HSCT) is a well-established curative therapy for patients with sickle cell disease (SCD) when using a human leukocyte antigen (HLA)-matched sibling donor. Most patients with SCD do not have a matched sibling donor, thereby significantly limiting the accessibility of this curative option to most patients. HLA-haploidentical HSCT with post-transplant cyclophosphamide expands the donor pool, with current approaches now demonstrating high overall survival, reduced toxicity, and an effective reduction in acute and chronic graft-vs.-host disease (GvHD). Alternatively, autologous genetic therapies appear promising and have the potential to overcome significant barriers associated with allogeneic HSCT, such as donor availability and GvHD. Here the authors each take a viewpoint and discuss what will be the future of curative options for patients with SCD outside of a matched sibling transplantation, specifically haploidentical HSCT vs. gene therapy.

## 1. Introduction

Sickle cell disease (SCD) is the most common inherited hemoglobinopathy worldwide and is associated with substantial morbidity and premature mortality [1,2]. Despite a significant increase in survival through newborn screening, penicillin prophylaxis, vaccinations, including use of disease-modifying therapies, such as hydroxyurea or chronic blood transfusions, they do not fully eliminate disease manifestations and require lifelong administration. Allogeneic hematopoietic stem cell transplantation (HSCT) is the only currently validated curative option for patients with SCD when using a human leukocyte antigen (HLA)-matched sibling donor and currently demonstrates an overall and event-free survival of >90% [3]. However, broad use of this option is significantly limited by donor availability, transplant-related morbidity and mortality, graft rejection, graft-versus-host disease (GvHD), infertility, and other treatment-related late-effects [4,5]. HLA-haploidentical transplantation for patients with SCD expands the donor pool for patients and now demonstrates high overall survival (OS), reduced toxicity, and an effective reduction in acute and chronic GVHD [6,7,8,9,10,11,12]. Simultaneously, significant advances in gene therapy suggest that an additional curative option(s) for patients might soon be available that will overcome significant barriers associated with allogeneic HSCT [13]. Here the authors each debate their viewpoint on the future of curative therapy for patients with SCD, including those without an HLA-matched sibling donor, taking into consideration patient eligibility, conditioning regimens, morbidity/mortality, efficacy, safety/risks, cost/accessibility, and patient preference.

## 2. The Case for Haploidentical HSCT as the Future of Curative Therapy for Patients with SCD

### 2.1. Haploidentical HSCT—Patient Eligibility

Improvement in transplant technology, supportive care, and increasing donor availability have increased access to this validated curative option for individuals with SCD [14,15]. Though children with SCD in high-income countries who receive high-quality supportive care no longer have a high rate of childhood morbidity and mortality [16,17,18], some continue to experience significant morbidities despite optimal medical therapy. Currently, accepted candidates for curative therapy are children with recurrent strokes (overt and silent), recurrent severe pain and acute chest syndrome despite optimal supportive care, pulmonary hypertension, high blood pressure, recurrent priapism, and complications associated with long-term poor outcomes [19,20,21,22,23,24]. In adults with SCD, HSCT may be an alternative to medical management given the high mortality rate [1,25]. Despite advances in medical therapy, the median survival for adults has remained relatively unchanged. The average life expectancy is 48 years in individuals with severe phenotypes (HbSS/HbSβ^o^/HbSD) and 54.7 years in those with milder phenotypes (HbSC/HbSβ^+^), which is on average ≥ 20 years shorter than the life expectancy for African Americans living in the United States without SCD [26]. Clustering of end-organ complications, such as stroke, chronic kidney disease, and cardiopulmonary disease, significantly contribute to morbidity and early mortality despite optimal medical therapies [27,28,29].

To be effective, a curative therapy must be applicable across ages and over a wide range of disease phenotypes. Matched sibling donor HSCT for SCD has shown excellent results in children over the last 3 decades when used in combination with myeloablative conditioning [3,14]. However, its application is limited by the availability of a suitable HLA-matched sibling donor (10–15%) [4,5], the toxicity of this approach (including infertility), GvHD, late effects, and a lack of awareness of the benefits of transplantation. Haploidentical HSCT (haplo-HSCT) using nonmyeloablative (NMA) conditioning has evolved as an alternative curative approach that addresses some of these unmet needs [30]. Most patients have a partially matched, or haploidentical, unaffected family member who could serve as a donor. Biological parents and children will share one HLA haplotype with a patient; 25% of siblings will be full HLA matches, and an additional 50% will share one haplotype; second-degree relatives (grandparents, grandchildren, nieces, nephews, and biological aunts and uncles) have a 50% chance of being HLA haploidentical, and first cousins will share an HLA haplotype 25% of the time. The ability to use such donors safely and effectively would make haplo-HSCT available to almost anyone, especially ethnic groups that are currently under-represented in unrelated donor registries [31]. The NMA approach has allowed for application across all ages and those with more severe diseases with minimal toxicity.

### 2.2. Haploidentical HSCT—Conditioning/Toxicity

Current haplo-HSCT approaches utilize either T-cell deplete (TCD) and unmanipulated, T-cell replete (TCR) approaches, utilizing sophisticated methods for graft manipulation in the former and drug-induced immunologic tolerance in the latter to allow for engraftment while minimizing GvHD [12,32,33,34,35,36]. Ex vivo TCD can broadly be divided into one of two strategies: CD34^+^-positive selection and CD34^+^-negative selection of T-cells [37]. These techniques differ not only in the laboratory procedure but also in the composition of the product, with CD34^+^ selection eliminating other populations of mononuclear cells, such as B- and NK cells, that can impact post-transplant immune reconstitution. TCR haplo-HSCT relies on in vivo strategies to overcome any HLA disparity and subsequent bi-directional alloreactivity [38,39,40].

The use of post-transplant cyclophosphamide (PTCy) has transformed the field of haplo-HSCT by allowing for selective deletion of alloreactive T-cells in vivo, which lack the expression of the drug-metabolizing enzyme called aldehyde dehydrogenase. PTCy modulates the alloreactivity associated with partially matched donors in animals and humans. Preferential expansion of regulatory T-cells may also contribute to the reduced GvHD seen with the PTCy approach [41,42,43,44,45]. Unlike gene therapy approaches, which utilize myeloablative conditioning to optimize marrow repopulation with genetically modified autologous cells, current haplo-HSCT protocols utilize novel NMA conditioning platforms to address the toxicity associated with prior HSCT approaches [11,38,39,46,47]. This has allowed for the extension of this curative approach to individuals with severe SCD (pediatric and adults) that were initially deemed high risk [11]. Some toxicities experienced with current myeloablative gene therapy approaches include grade ≥ 3 cytopenia, sepsis in presence of neutropenia, and transaminitis [13,48]. Table 1 highlights the pros and cons of differing approaches for haplo-HSCT.

### 2.3. Haploidentical HSCT—Morbidity/Mortality

Most adults with SCD have significant and sometimes overlapping comorbidities, such as strokes (silent or overt) or significant heart, lung, or kidney disease, thus are not eligible for curative therapy trials [28]. Rapid immune recovery post-HSCT is important to avoid short- and long-term infectious complications. Excellent immune reconstitution was achieved following TCD haplo-HSCT with mononuclear cell addback, with the potential to obviate delayed engraftment and increased opportunistic infections with TCD-associated delayed immune reconstitution [49]. In a cohort of 23 patients with severe SCD, Patel et al. showed rapid immune reconstitution in B, T, and NK cell subsets post-transplant in all patients following haplo-HSCT using TCR bone marrow grafts and post-transplant cyclophosphamide [50]. Infections due to viral reactivation caused by delayed or impaired T-cell mediated immunity are the major complications of TCD and TCR haplo-HSCT [11,51]. Cytomegalovirus, Epstein–Barr virus, human herpesvirus 6, and adenovirus reactivation were common, and BK-associated hemorrhagic cystitis occurred in 35% of patients [51]. In addition, bacterial and fungal infections were also noted and contributed to the morbidity of this approach [52]. The current platform using TCR, nonmyeloablative haplo-BMT with PTCy, was not associated with increased transplant-related mortality [11]. Two patients had grades III–IV acute GvHD, one patient had mild chronic GvHD, and 86% (6/7) of patients were off all immunosuppression by 1-year post-transplant.

When comparing haplo-HSCT to the alternative discussed here (gene therapy), most participants described in the largest reported gene therapy study to date [13] had no documented major organ dysfunction—no cases of stroke, pulmonary hypertension, or chronic kidney disease. These significant complications were prevalent in current haplo-HSCT cohorts [7,11], which raises the question of the applicability of gene therapy to older adults with significant co-morbidities. Table 2 highlights the complications of the two curative strategies for SCD, haplo-HSCT, and gene therapy.

### 2.4. Haploidentical HSCT—Efficacy

Successful HSCT results in the reversal of all SCD-related clinical symptoms and improvement in chronic organ dysfunction [14,30]. Current TCD and TCR-based approaches for haplo-HSCT increase donor availability and demonstrate reduced regimen-related toxicity and rates of GvHD, with higher engraftment rates. Promising transplant outcomes were demonstrated when using more refined selective means of TCD grafts [51,53]. A phase II study in children and young adults with severe SCD investigated the use of NMA conditioning with CD34^+^-selected peripheral blood TCD grafts to minimize GvHD and the risk of rejection [53]. Eight out of ten patients with a median age of 14 years (range: 5–23 years) underwent haplo-HSCT. All eight patients engrafted with the incidence of grades II–IV acute GvHD reported in 20% of the patients. There was one case of chronic GvHD, which occurred in a patient with previous acute GvHD who also received a donor lymphocyte infusion for refractory post-transplant lymphoproliferative disorder (PTLD). At two years, the overall survival was 90% and the event-free survival was 80%, suggesting that outcomes with NMA are improving with the TCD haplo-HSCT platform. Gaziev and colleagues published the result of a single-institution, retrospective study of children and young adults with hemoglobinopathies who underwent myeloablative conditioning, followed by selective depletion of TCR alpha/beta^+^ cells [51]. The median age was 7 years (range 3–15.2). They compared their outcomes to a group of 40 patients with hemoglobinopathies who received CD34^+^-selected peripheral blood and bone marrow grafts (*n* = 32) or CD34^+^-selected peripheral blood and CD3^+^/CD19^+^-depleted bone marrow grafts (*n* = 8). Both groups were similar in baseline characteristics and the 5-year probability of overall survival (OS) and disease-free survival (DFS) was 84% and 69%, respectively, for the T-cell receptor alpha/beta^+^ and CD19^+^ group compared with 78% and 39%, respectively, for the CD34^+^-selected control group. Graft failure was significantly lower in the T-cell alpha/beta^+^ and CD19^+^ group compared with the historical cohort (14% vs. 45%, *p* = 0.048) (Table 2). Outcomes in terms of acute GvHD, chronic GvHD, PTLD, and viral reactivation were not significantly different between the groups.

The use of haplo-HSCT in adults with SCD was first described using a nonmyeloablative regimen of anti-thymocyte globulin, fludarabine, and total body irradiation (TBI), followed by a bone marrow graft (haplo-BMT) [31]. PTCy, Mycophenolate, and Sirolimus were used for GvHD prophylaxis. The median age was 23.5 years. At a median follow-up of 711 days, engraftment occurred in 57% (8/14) of the recipients, with graft rejection occurring in the remaining 43% (6/14) of patients. All patients who lost their graft recovered autologous hematopoiesis. All engrafted patients had no acute or chronic GvHD and 100% OS. This regimen has since been improved with increased TBI dosing to 400 cGy, substantially reducing graft failure [6]. Other investigators have tried to address this unmet need of reducing the graft rejection rate using various strategies, including granulocyte colony-stimulating factor bone marrow priming to increase the T-cell content in the graft [54] while reducing donor-specific HLA antibodies, which are associated with high graft rejection rates in haplo-HSCT recipients [55,56]. The NIH group developed a novel nonmyeloablative HLA-haploidentical peripheral blood stem cell transplant approach that could safely be used for patients with severe organ damage using low-dose TBI and alemtuzumab followed by escalating doses of PTCy: 0 mg/kg in cohort 1, 50 mg/kg in cohort 2, and 100 mg/kg in cohort 3 [7]. Of the initial 21 patients transplanted, 50% (6/12) had graft rejection in cohort III (in patients who received the highest doses of PTCy), 86% survived, and three patients died, all of which occurred in patients who had graft rejection.

Recently, a multi-institutional phase II study of haplo-BMT using non-myeloablative conditioning and PTCy for patients with severe SCD was published [11]. A total of 16 patients underwent 18 haplo-HSCT, where the median age was 20.9 years, and the first three received conditioning similar to the Johns Hopkins platform [31]. However, two-thirds experienced graft rejection, which necessitated the addition of thiotepa to the conditioning. Most patients received granulocyte colony-stimulating factor primed bone marrow grafts. Ninety-three percent of patients (14/15) who received the new regimen, including two who had a previous graft rejection, had stable myeloid donor engraftment after at least 6 months of follow-up. No mortality was seen, two patients had grades III–V acute GvHD, one had mild chronic GvHD, and 86% (6/7) of patients were off all immunosuppression by 1-year post-transplant. Importantly, the results suggest that the addition of thiotepa to the Johns Hopkins haplo-HSCT with the PTCy platform may be an effective strategy to improve engraftment, potentially extending this curative modality to individuals with severe SCD, including older patients. This approach formed the basis for an on-going multicenter national study in the United States, BMT CTN Protocol 1507 (clinicaltrials.gov as NCT03263559). The last few years have seen incremental improvements in transplant technology, with improved outcomes and efficacy using this alternative donor approach. Table 3 reviews transplant outcomes from published studies using TCD and TCR platforms for haplo-HSCT for SCD.

### 2.5. Haploidentical HSCT—Safety/Risk

Current haplo-HSCT platforms are exploring the use of NMA approaches with PTCy as a means of decreasing toxicity, improving engraftment, and minimizing GvHD while maintaining efficacy in children and adults with severe SCD [11,31,52,57]. Similarly, TCD grafts demonstrate high OS and EFS [51,53], and TCR-based approaches show improved safety in participants despite the increasing disease severity of participants [7,11]. In the haplo-BMT study by de la Fuente et al., although most participants had significant, sometimes overlapping co-morbidities (>80% had recurrent acute chest syndrome, 75% had cerebrovascular disease (overt stroke and silent cerebral infarcts), and 56% had frequent acute vaso-occlusive pain episodes despite hydroxyurea therapy), the conditioning was well tolerated. At a median follow-up of 13.3 months, 93% had stable donor engraftment with 100% OS. Overall, current results suggest an improved safety profile with NMA haplo-HSCT platforms despite the disease severity and support the generalizability of this approach.

Ghannam and colleagues reported on the development of myeloid malignancy in three individuals with homozygous SCD occurring about 2–5 years after failed allogeneic HSCT [58]. Two of these participants had baseline TP53 mutations or clonal hematopoiesis of indeterminate potential (CHIP) prior to HSCT. SCD is recognized to have a low absolute but increased relative risk for hematologic malignancies [59]. Plausible underlying mechanisms implicated include chronic hypoxia, endothelial damage, chronic systemic inflammation, disease-related immunomodulation, and erythropoietic stress with dysregulated apoptosis [60,61,62,63] (Figure 1). In a large retrospective multicenter, cohort study that investigated the effect of donor type and conditioning regimen intensity on allogeneic transplantation outcomes in patients with SCD, 6 (1%) out of 910 patients developed malignant neoplasm post-transplantation. Further understanding of the predisposing factors to secondary malignancy in patients with SCD following HSCT is needed. Table 2 reviews the toxicities and complications of haplo-HSCT.

### 2.6. Haplo—Cost/Accessibility

SCD-related complications, particularly recurrent pain episodes, are associated with high healthcare costs, resulting in high financial burdens. Total healthcare costs are estimated to increase with age from USD 892 per month in the <10 age group to USD 2853 per month in the 30–39 age group [64]. The nonelderly (0–64 years of age) lifetime burden of total medical costs attributable to SCD was USD 1.7 million [65]. Patients incurred USD 44,000 in out-of-pocket costs due to SCD over their nonelderly lifetimes. A successful outcome after curative therapy for SCD early in life could have a significant beneficial effect on lifespan and the quality of life, as well as reduce life-long healthcare expenditures.

HSCT carries a significant financial cost in the first year. The reported cost of HSCT for adults with malignant or nonmalignant conditions in the first-year ranges from USD 96,000 to USD 204,000. This cost can vary based on the conditioning regimen, allograft type, and donor source. In contrast, the median estimated transplant cost in children during the transplant year is approximately USD 413,000 per patient, suggesting that factors unique to pediatric populations confer increased costs compared with adults [66]. Despite high upfront costs, healthcare utilization decreases significantly over time post-HSCT when compared with that pre-HSCT and with control subjects. More importantly, this change is associated with significant health-related quality of life (QOL) improvements. However, this cost can increase with high-grade acute GvHD, infectious complications, and unrelated donor transplant complications, which are being addressed with current novel approaches to haplo-HSCT. The expensive graft manipulation technology used in the TCD-based approach pits itself against the relatively inexpensive drug-induced immunologic tolerance used in the TCR-based approach with PTCy.

### 2.7. Haplo—Patient Preference

The current approach for NMA haplo-BMT was developed to increase donor availability, minimize GvHD, and reduce toxicity. A survey on the patient perception of reduced-intensity transplantation in adults with SCD suggested that the majority of adults with SCD might be willing to consider a reduced intensity curative HSCT option even with a high treatment-related mortality or graft failure [67]. The major concerns relate to chronic GvHD and infertility, which have improved with current NMA approaches for HSCT and PTCy as GvHD prophylaxis. Although the success rate of 50% of the NIH haplo-HSCT regimen is not ideal, it is important to interpret these results in the context of the very severe disease phenotype and comorbidities of the patients treated in this study. The authors reported that 50 out of 100 patients, including those with significant organ damage, could potentially be cured compared with 14 of 100 patients in the HLA-matched sibling setting in which the success rate is closer to 90% but the chance of having an HLA-matched sibling is about 14% [7]. In this context, haplo-HSCT is preferable to current gene therapy approaches, which currently utilize myeloablative conditioning to maximize the engraftment of gene-modified cells, froth with unknown risks and yet to be defined long-term complications.

## 3. The Case for Gene Therapy as the Future of Curative Therapy for Patients with SCD

### 3.1. Gene Therapy—Patient Eligibility

Transplantation with gene-modified autologous HSCs is theoretically available to all patients who would otherwise qualify for transplantation given that each patient serves as their own donor. Less than 15% of patients have an HLA-matched sibling donor [4,5,68], and while alternative donor sources, such as HLA-haploidentical HSCT, offer more patients the chance for cure, the risk of rejection, GvHD, and the need for chronic immune suppression are significant limitations that are overcome in the autologous setting.

Given the experimental nature of gene therapy, gene therapy trials are currently limited to patients who meet strict inclusionary criteria, which are typically characterized by severe, recurrent vaso-occlusive events (VOEs). Furthermore, patients with an available HLA-matched sibling donor are excluded from pursuing gene therapy as a curative option given favorable outcomes after allogeneic HSCT with a matched related donor [3,69,70]. Patients who have had prior receipt of allogeneic transplantation are not eligible for gene therapy, in part due to the requirement for autologous stem cell collection. It is important to note that the strict inclusionary criteria were designed to allow for evaluation of the number of patients who have complete resolution of severe vaso-occlusive events as an efficacy endpoint. As data for gene therapy becomes available with longer follow-ups and as companies seek FDA approval for genetic technology for the cure of hemoglobinopathies, inclusionary and exclusionary criteria will likely change and more closely resemble that of allogeneic HSCT.

For a patient who is otherwise eligible, and interested in a curative option, without a matched related donor, gene therapy is not an option if the patient chooses to undergo haploidentical HSCT that results in graft rejection or malignant transformation. Alternatively, a patient can undergo haploidentical HSCT after gene therapy if there is a suboptimal clinical benefit or malignant transformation post-gene therapy [71].

### 3.2. Gene Therapy—Conditioning/Toxicity

Current gene therapy protocols require myeloablative conditioning to maximize marrow repopulation with genetically modified autologous cells (exception, NCT02186418). Therefore, the use of myeloablative conditioning restricts the broad use of gene therapy to those who can safely tolerate myeloablative chemotherapy, denying patients who may otherwise qualify for transplantation but have substantial comorbidities that bar them from safely undergoing myeloablation. In the allogeneic setting, 20–25% donor myeloid chimerism is sufficient to reverse the sickle phenotype due to the survival advantage of donor vs. recipient red blood cells [72,73]; therefore, reduced intensity conditioning is possible and may be preferred, though early and late graft failures remain a challenge. The amount of additional/corrected globin or the amount of fetal hemoglobin expression from genetically modified hematopoietic stem cells (HSCs) needed to correct the disease phenotype is currently unknown; therefore, future trials in gene therapy may be able to tolerate non-myeloablative conditioning if the target vector copy number or editing rates that restore red blood cell lifespan are better understood.

However, the future of conditioning appears to be promising for antibody-based conditioning regimens. Antibodies to CD117 (c-kit) selectively target HSCs and, therefore, have the potential to offer less toxic conditioning with an improved risk–benefit profile in the autologous setting. The selective depletion of HSCs allows for immune and fertility preservation and reduces the risk of treatment-associated malignancy that is often associated with the use of chemotherapy. Early mouse models demonstrated improved safety profiles with comparable efficacy to standard conditioning regimens while also demonstrating preserved immune function [74,75]. Recently, a single dose of antibody-drug conjugate (ADC) CD117-ADC allowed for efficient engraftment of gene-modified CD34^+^ HSCs in a rhesus gene therapy model, achieving a similar level to myeloablative busulfan conditioning [76]. Antibody-based conditioning may be useful in the allogeneic setting but is likely to be required in synergy with other modalities to achieve both myelodepletion and immune suppression.

### 3.3. Gene Therapy—Morbidity/Mortality

Graft rejection, acute and/or chronic GvHD, and infectious complications from delayed immune reconstitution are major sources of morbidity and mortality following alternative donor allogeneic HSCT for SCD. In its current form, allogeneic HSCT and autologous HSCT demonstrate a common risk of toxicity associated with conditioning reagents, such as busulfan; however, by eliminating any risk of GvHD or rejection, gene therapy eliminates major sources of morbidity and mortality and, therefore, is preferable to allogeneic transplantation. To date, there are no reports of GvHD, immune rejection, veno-occlusive liver disease, circulating replication-competent lentivirus, clonal dominance, or insertional oncogenesis. The overall safety profile is generally consistent with that of myeloablative conditioning and that of the underlying disease. In the largest gene therapy trial to date, the median time until neutrophil engraftment was 20 days (range, 12 to 35), and the median time until platelet engraftment was 36 days (range, 18 to 136) [13]. Serious adverse events were consistent with myeloablative conditioning or possibly attributed to Lentiglobin infusion (grade 2 leukopenia, grade 1 decreased diastolic blood pressure, grade 2 febrile neutropenia), and all were resolved within 1 week after onset.

Since the first clinical trial opened for gene therapy in SCD in 2013, there have been three reported deaths following gene therapy for SCD. One death occurred 20 months after infusion in a patient with significant baseline SCD-related cardiopulmonary disease and was, therefore, deemed unrelated to the LentiGlobin infusion [13]. Two patients developed acute myeloid leukemia (AML) three and five years after gene therapy, prior to significant changes to the trial design, which included improvements in stem cell collection and the drug manufacturing process [77]. One individual developed myelodysplastic syndrome (MDS) three years after treatment, which eventually transformed into AML. The absence of a vector among the blasts, along with complex cytogenetic abnormalities and driver gene mutations, suggested that this complication arose from the busulfan conditioning and was unrelated to the lentiviral vector [71]. The second individual had AML blast cells that contained a BB305 lentiviral vector insertion site that was determined to be in *VAMP4*, which is a gene with no reported role in oncogenesis. This insertion site had no significant effects on gene expression, indicating that vector insertion was a passenger to AML development and not a driver of oncogenesis. There have been no cases of hematologic cancer observed for up to 37.6 months of follow-up in the most recent version of this same trial.

### 3.4. Gene Therapy—Efficacy

As of February 2022, more than 50 individuals with SCD had received gene therapy, all of which demonstrated engraftment of the gene-modified cells. The initial bluebird bio gene addition study (cohort A) demonstrated a generally low peripheral blood vector copy number (VCN) and low modified globin expression [78]. Improvements were subsequently made across the design of the study: a transition from the use of bone marrow as the source of HSCs to plerixafor mobilized HSCs, improved standards for refined manufacturing, and the requirement of pre-harvest transfusions [79,80,81,82,83,84]. Many of these modifications are now standard across gene therapy clinical trial designs. As of February 2022, there were eight clinical trials listed on clinicaltrials.gov using gene addition therapy for the treatment of SCD and six clinical trials using gene editing for the treatment of SCD using either zinc finger nuclease (ZFN) products or CRISPR/Cas9 products.

Available data suggest that after implementation of the use of plerixafor-mobilized HSCs, refined manufacturing, and pre-harvest transfusions, the current gene therapy products are associated with 100% engraftment of gene-modified cells, high levels of modified gene expression, and disease amelioration. The most studied gene therapy product in development for SCD, namely, LentiGlobin BB305 (bb1111, lovotibeglogene autotemcel), recently reported clinical trial data that showed engraftment in 35 of 35 patients, a median total hemoglobin level increase from 8.5 g/dL at baseline to ≥11/dL, and modified hemoglobin (HbA^T87Q^) that contributed at least 40% of total hemoglobin distributed across 85% of red cells [13]. Among the 25 patients who could be evaluated, all displayed resolution of severe vaso-occlusive events (VOEs) as compared with a median of 3.5 events per year (range, 2.0 to 13.5) in the 24 months before enrollment. In addition, there was an improvement in and a sustained and clinically meaningful quality of life benefit for the patients [85].

Several other gene addition trials reported on small numbers of patients treated and include gene delivery of a modified gamma globin gene, ARU-1801 (NCT02186418), erythroid-specific expression of a short hairpin RNA targeting *BCL11A* (NCT03282656), or delivery of anti-sickling globin (NCT03964792). Four patients were treated with ARU-1801, of which the total fetal hemoglobin (HbF) percentage was 15–37% with a VOE reduction percentage of 80–100% [86,87]. Initial results in six patients after genetic silencing of *BCL11A* demonstrated a median percentage of F-cells (cells containing HbF) of 70.8%, which was a robust increase from a median of 14% at baseline, and significant attenuation of the acute sickling phenotype [88]. Results from three patients with 8–18 months of follow-up after Drepaglobe were mixed, demonstrating 20–30% of modified globin expression with ongoing hospitalizations due to vaso-occlusive crisis or ongoing transfusion requirements [86].

Data from gene editing trials are generally limited in terms of patient numbers and the length of follow-up. SAR445136 (formerly BIVV003) used ZFNs to disrupt the *BCL11A* erythroid enhancer and reported data in four patients with follow-ups ranging from 26–91 weeks (NCT03653247) [89]. On-target editing resulted in 61–78% INDELs in the drug products, 25% INDELs in the bone marrow, and HbF levels ranging from 14 to 38%. CTX001 used CRISPR/Cas9 to disrupt the BCL11A erythroid enhancer to increase endogenous HbF (NCT03745287) and reported that HbF comprised 45% of total hemoglobin 6 months post-treatment in five patients with a median total hemoglobin of 13.7 (11–15.9) g/dL with 96% F+ cells reported in four patients [48,90]. Table 4 reviews different gene therapy approaches.

### 3.5. Gene Therapy—Safety/Risk

The safety profile post-gene therapy remains generally consistent with the risk of autologous stem cell transplantation, myeloablative busulfan conditioning, and underlying SCD. Available data suggest efficient transduction or INDEL formation in SCD HSCs, no adverse events related to genetic technology, lineage-specific expression, stable gene marking over time, and clinically meaningful SCD symptom resolution. While early clinical data are promising, the small sample sizes and limited clinical follow-up warrants caution in data interpretation, particularly as the two cases of hematologic malignancy were only evident with time.

The risks associated with genetic manipulation are significant. Lentiviral addition strategies require integration into the host genome; therefore, thousands of insertional mutations occur in a population of treated cells. In 30 patients with available data, no unique insertion site was present at more than 3.8% of all unique insertion sites at any time point [13]. Though CRISPR is non-integrating, off-target CRISPR-induced DNA modifications are potentially deleterious and double-stranded DNA breaks may reduce engraftment and proliferative capacity such that a population becomes selected for with proliferative advantages.

There are several potential predisposing factors for myeloid dysplasias following HSCT for SCD. Patients with SCD have an increased relative but a low absolute risk of AML/MDS at baseline [59]. Pre-existing clonal hematopoiesis of indeterminant potential (CHIP)-related mutations compound the already known risks associated with genotoxic conditioning [58]. Allogeneic and autologous HSCT by itself carries a known risk of therapy-related myeloid neoplasms [70], and the presence of CHIP at the time of transplant raises the 10-year cumulative incidence [91]. The two patients who developed AML following gene therapy had an inadequate therapeutic response that was consistent with data suggesting MDS/AML following allogeneic HSCT is more common in patients with graft failure. Current theories suggest that after graft rejection or inadequate therapeutic expression of the gene of interest after autologous gene therapy, the stress of switching from homeostatic to regenerative hematopoiesis using autologous cells drives clonal expansion and the leukomogenic transformation of pre-existing premalignant clones, eventually resulting in AML/MDS [92]. Because stem cell abnormalities may be present or enhanced by the gene modification process, ongoing efforts are being implemented to screen for existing MDS features or pathogenic mutations associated with hematologic malignancies prior to genetic manipulation of autologous HSCs all while continuing to improve upon transgene expression.

While the risks for genetic manipulation are significant, current data suggest that underlying SCD, transplant conditioning, and rejection rather than the genetic technology per se are responsible for current safety concerns. When all else is equal, when engraftment is 100% with high transgene expression after gene therapy, the risk of rejection and GVHD after haploidentical transplantation is therefore of significantly higher risk to patients.

### 3.6. Gene Therapy—Cost/Accessibility

SCD is a chronic disease with high healthcare utilization; therefore, management becomes more costly over time. For a 24-year-old patient on usual care, the lifetime total cost is estimated to be USD 1.1 million, exceeding the USD 8 million for a patient surviving to age 50 [93,94]. Therefore, curative approaches represent a suitable strategy to reduce personal lifetime healthcare costs and reduce hidden costs that are often not factored into healthcare estimates, including the loss of wages due to frequent healthcare visits, unemployment among patients and parents, and reduced quality of life (QOL).

Curative therapy is likely to be cost-effective given large upfront costs are offset by significant downstream gains in health for patients treated early in life. In probabilistic sensitivity analysis, durable treatment is cost-effective at a minimum willingness to pay (WTP) of USD 150,000 per quality-adjusted life year (QALY) at single administration costs of USD 2.18 M for a cure duration lasting a lifetime [95]. Curative therapy generates an average of 26.4 discounted QALYs at a cost of USD 2,372,482 per patient vs. standard of care resulting in 17.9 discounted QALYs at a discounted cost of USD 1,175,566 per patient. Durable therapy results in an incremental cost-effectiveness ratio of USD 140,877 per QALY and is hence cost-effective at a WTP threshold of USD 150 K per QALY.

In addition to being comparable in cost to the standard of care, curative therapy appears to be more cost-effective than emerging disease-modifying therapies (voxelotor, crizanlizumab, l-glutamine). The median allogeneic HSCT cost per patient with SCD is estimated at USD 467,747 (range: USD 344,029–799,219) [96], whereas the lifetime treatment cost of crizanlizumab is USD 970,000, voxelotor is USD 1.1 million, and l-glutamine is approximately USD 299,000 [94]. Costs for gene therapy are less certain and suggested to be as high as USD 900,000–2.1 million [97], currently making gene therapy prohibitive as a realistic cure that is available to all. The additional costs for gene therapy occur largely in the pre-transplant period, yet costs appear homogeneous during the transplant process between patients treated by allogeneic HSCT or gene therapy and are significantly lower in the follow-up period [98]. Patients treated by gene therapy had lower costs in the follow-up period owing to fewer infectious complications, treatments, imaging, outpatient care, and inpatient admissions. Gene therapy patients had fewer productivity losses and experienced fewer complications, hospital admissions, and had shorter hospital stay lengths but cost an additional GBP 300,000–400,000 per patient on average. The additional costs associated with gene therapy could be justified due to better clinical effectiveness, less post-transplant ongoing care, and fewer costs associated with managing transplant comorbidities, such as GvHD or rejection.

Both allogeneic transplantation and gene therapy methods require specialized centers for patient care, currently limiting widespread accessibility. Pre-transplant donor availability and post-transplant follow-up are much less onerous in the autologous setting, and the future of gene therapy is likely to include antibody-based conditioning and/or in vivo gene delivery, giving a significant advantage to gene therapy methods over alternative allogeneic donor transplantation.

### 3.7. Gene Therapy—Patient Preference

Patients perceive HSCT as a treatment option associated with serious risks [99]. However, a substantial proportion of adults are interested in curative treatment, even at the expense of considerable risk. Nearly three-quarters of patients that are interested in curative therapy were willing to accept some short-term risk of mortality and more than 50% said they were willing to accept a risk of ≥10% of GvHD in exchange for the certainty of a cure [100,101]. Parents of children with SCD sought transplant consultation because of their child’s diminished QOL, recent complications, an imminent major medical decision, or anxiety about future severe complications [102]. Those same parents perceived gene therapy as a new, less invasive, and more acceptable treatment. Compared with allogeneic transplantation, care post gene therapy is far less complicated given no immune suppression and has no risk of rejection or GvHD, fewer complications and hospitalizations, and a shorter length of stay. Given the speed of recovery after transplantation and the immediate lack of sickle-related symptoms, the only transplantation debate left in SCD will not be haploidentical vs. gene therapy but rather which method of gene therapy is best.

## 4. Conclusions

SCD can be cured following HSCT and is a means not only to change the life of an individual living with SCD but also to reduce the growing burden of SCD worldwide. Current results of TCR haplo-HSCT are addressing the unmet needs of the established HSCT approach, namely, HLA-identical sibling HSCT for SCD, with increasing donor availability, reducing toxicity, and resulting in less chronic GvHD. Given improvements in transplant technology, recent data suggests haploidentical transplantation significantly widens donor availability with high overall and event-free survival. However, to become the gold-standard curative strategy for SCD, graft rejection will need to be addressed. Gene therapy appears to be effective and generally safe, though data is limited by small patient numbers and a lack of long-term follow-up. In contrast to haplo-HSCT, patients do not have to contend with substituting one chronic disease for another, do not need immune suppression, and experience rapid symptom relief without the typical post-transplant complications from allogeneic transplantation. As these two curative strategies continue to improve and data matures, the best outcome is that patients have multiple curative options from which to choose. Such a future places patients and their autonomy at the center of a safe and effective curative therapy.

## Figures and Tables

**Figure 1 jcm-11-04775-f001:**
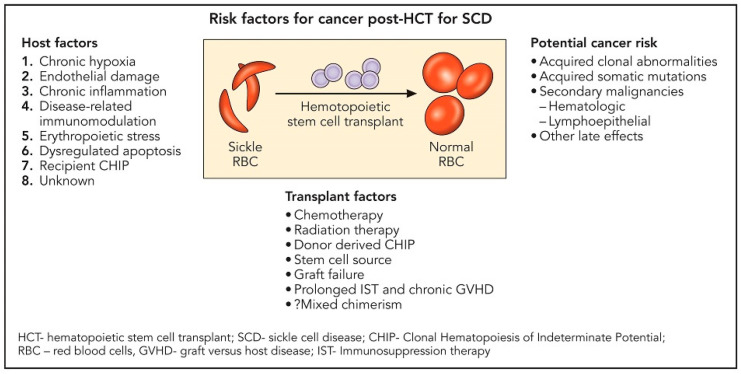
Risk factors for cancer post-transplantation for sickle cell disease.

**Table 1 jcm-11-04775-t001:** Pros and cons of commonly used TCD vs. TCR approaches used in haplo-HSCT.

T-Cell Deplete Method	Mechanism	Pros	Cons
General	Multiple (listed below)	Conceptually most effective means to prevent acute and chronic GvHD Low acute and chronic GvHD Reduced need for post-transplant immune-suppressive medications Lower pulmonary and hepatic toxicity peri-transplant Prevents EBV-PTLD (high potential morbidity and mortality) by removing CD19^+^ cells ex vivo	More effective in children than adults (due to better thymus function in children with associated greater T-cell receptor diversity versus adults, who rely more on peripheral cytokine-mediated T-cell expansion post-transplant) Expensive Labor-intensive Specialized expertise required Not available at most stem cell transplant centers Higher graft rejection/lower engraftment related to depletion of T-cells (especially gamma/delta), natural killer cells, and hematopoietic progenitors that facilitate engraftment Delayed IR with increased risk of opportunistic infections
CD34-positive selection	Positive selection of CD34^+^ stem cells via immunoadsorption columns (immunomagnetic beads) Combined physical and immunological separation of cells	Beneficial for engraftment (barrier overcome by “megadose” CD34^+^ stem cell infusion)	Loss of cells that facilitate engraftment, such as gamma/delta T-cells and natural killer cells, with a subsequent increased risk of graft rejection Potential for severely delayed IR with increased infectious risk profile for many months to years and conceptual risk of relapse of disease (return of sickle cell disease phenotype) Myeloablative conditioning is used more often (accentuates existing end-organ damage, higher risk of acute and chronic GvHD, higher transplant-related mortality)
CD3^+^ and CD19^+^	Ex vivo negative selection of CD3 (T-cells) and CD19 (B-cells)	Lower risk of EBV-PTLD (from removing potential EBV-infected CD19 cells in the graft)	Risks as described in “General” and “CD34 positive selection” Loss of cells that promote engraftment (gamma/delta T-cells and natural killer cells)
T-cell receptor alpha/beta^+^ and CD19^+^	Ex vivo depletion of more specific T-cell subsets that drive acute GvHD and B-cells that increase the risk of EBV-PTLD	Retain gamma/delta^+^ T-cells (promote IR and provide pathogen-specific immunity) and natural killer cells while depleting alloreactive T-cells that cause acute GvHD Less delayed T-cell specific IR	Requires even more specialized expertise than CD34-positive selection methods of CD3^+^ and CD19^+^ negative selection Available at fewer centers Data only reported in children
**T-Cell Replete Method**	**Mechanism**	**Pros**	**Cons**
General	In vivo rather than ex vivo depletion of recipient and donor alloreactive T-cells (with anti-thymocyte globulin or alemtuzumab, with or without total body or lymphoid irradiation)	Available at almost all transplant centers in Europe and the United States Methods are more easily replicable Conceptually lower cost compared with T-cell depletion methods (due to the lack of a need for expensive graft-manipulation technology)	Need for in vivo T-cell depletion with anti-thymocyte globulin or alemtuzumab, with potential for delayed IR and increased risk of opportunistic pathogens Higher GvHD risk with peripheral blood stem cell grafts Potential for severe cytokine release syndrome (especially with peripheral blood stem cell grafts) due to rapid activation of T-cells
GIAC protocol	Modulation of alloreactive T-cells with (1) Granulocyte colony-stimulating factor donor priming, (2) Intensive immunosuppression post-transplant, (3) Anti-thymocyte globulin, (4) Combined peripheral blood and bone marrow allografts	Reduce alloreactivity of donor T-cells with granulocyte colony-stimulating factor (shift from T-helper 1 to T-helper 2 phenotype) and of both donor and recipient T-cells with anti-thymocyte globulin Improved engraftment due to the use of peripheral blood stem cells No need for graft manipulation Protocols are easily replicable at different institutions	Morbidity from multiple drugs needed for post-transplant immune Increased risk of viral reactivation and opportunistic pathogens in the early post-transplant period due to anti-thymocyte globulin Unanswered question regarding non-inherited maternal and paternal antigens (for donor selection) Not as extensively studied in the setting of SCD
Post-transplant cyclophosphamide	Preferential deletion of proliferative alloreactive donor and recipient T-cells due to lack of expression of the enzyme aldehyde dehydrogenase 1 Reduce host T-cells responding to donor antigens peripherally post-transplant Intrathymic deletion of donor-reactive host T-cells (central tolerance)	Reduced acute and chronic GvHD Expansion of regulatory T-cells that promote immune tolerance Replicable at any transplant center Used with either bone marrow or peripheral blood stem cell grafts (compared with GIAC protocol) Low likelihood of developing EBV-PTLD Low documented incidence of donor-derived malignancies	Graft rejection chance is high (Bolanos-Meade et al. [31]) with the Johns Hopkins protocol alone, but is improved with the addition of thiotepa Potential acute toxicity from high doses of cyclophosphamide, including cardiac (type I agent, with hemorrhagic necrosis and heart failure), lung (pneumonitis and pulmonary fibrosis), bladder (associated with BK virus cystitis), secondary malignancy (chromosome 5 and 7 deletion signature, from alkylating agent exposure) Increased viral reactivation Increased risk of infertility secondary to additional alkylator therapy

Legend: haplo-HSCT, haploidentical hematopoietic stem cell transplant; TCD, T-cell deplete; EBV, Epstein–Barr virus; PTLD, post-transplant lymphoproliferative disorder; IR, immune reconstitution; TCR, T-cell replete; GvHD, graft-versus-host disease.

**Table 2 jcm-11-04775-t002:** Comparison of the two curative therapies for adults with severe SCD.

Variables	Haploidentical BMT with Post-Transplant Cyclophosphamide	Current Gene Therapy Approaches
Curative	Yes	Yet to be validated
Intensity of regimen	Non-myeloablative	Myeloablative
Eligibility	Most adults with organ dysfunction	Limited to children with no organ dysfunction
Donor availability	>90% will have eligible related haploidentical donors	None needed (autologous)
Stem cell procurement	Single bone marrow harvest or peripheral stem cell mobilization of eligible family donor	Requires multiple apheresis cycles
Toxicity of regimen	High-dose Cytoxan short-term toxicity (hemorrhagic cystitis, cardiotoxicity, pulmonary fibrosis, immunosuppression, increased hepatic enzymes and syndrome of inappropriate anti-diuretic hormone (SIADH), which is limited with supportive care.	High-dose busulfan toxicity (short-term—seizures, cardiovascular, gastrointestinal, bronchopulmonary dysplasia with pulmonary fibrosis and hepatic sinusoidal obstruction syndrome).
Outcomes	Evidence that a successful transplant attenuates progressive vasculopathy and end-organ damage	Unknown impact on progressive vasculopathy and end-organ damage in adults
Complications	Risk of GVHD and graft rejection	Avoids immunologic complications (GVHD or graft rejection) Poor phenotypic correction Poor consistency, integration, and site-independence Poor-level expression of the inserted gene Poor erythroid lineage specificity; developmental stage-specific expression of the inserted gene.
Late-effects	Long-term—less risk of ovarian failure, puberty, amenorrhea, or development of myeloid disorders from recipient derived clonal hematopoiesis of indeterminate potential (CHIP) in engrafted patients with current NMA approaches.	Long-term—ovarian failure, failure to achieve puberty and amenorrhea, secondary malignancies with current myeloablative conditioning with Busulfan. Chromosomal alterations may also occur; possible genotoxic effects; creation of DSBs at locations other than the desired genomic location; risk of clonal hematopoiesis of indeterminate potential (CHIP) prior to HSCT
Requirements	Requires only a FACT-accredited facility	Requires both GMP and FACT accredited facilities

Legend: GVHD, graft-versus-host disease; SCD, sickle cell disease; HSPCs, hematopoietic stem and progenitor cells; iPSCs, induced pluripotent stem cells; DSBs, double-strand breaks; GMP, good manufacturing practice; FACT, Foundation for the Accreditation of Cellular Therapy; HSCT, hematopoietic stem cell transplant.

**Table 3 jcm-11-04775-t003:** Transplant outcomes from published studies using TCD and TCR platforms for haplo-HSCT for SCD.

Author	Graft Source	Conditioning Regimen	N	OS	GvHD	Engraftment (%)	Complications
Gaziev et al. [51] T-cell receptor alpha/beta^+^ CD19 depletion	PBSC	Hydroxyurea and azathioprine with fludarabine pre-conditioning ATG, busulfan, thiotepa, and cyclophosphamide GvHD prophylaxis—cyclosporine and methylprednisolone or cyclosporine and MMF	3 sickle cell disease and 11 thalassemia	84% at 5 years	36% (5/14) acute GvHD 21% (3/14) chronic GvHD	86%	4 developed auto-immune disorders Infections Reactivation of CMV and EBV BK virus Adenovirus Bacterial infections with Gram-positive and Gram-negative sepsis Fungal
Gilman et al. [53] CD34^+^ selection	PBSC	Reduced intensity ATG, melphalan, thiotepa, fludarabine GvHD prophylaxis—none	8	88% (7/8) at over a range of 6–60 months	25% (2/8) grades II–IV acute GvHD 12.5% (1/8) moderate–severe chronic GvHD	100% (8/8)	2 with engraftment syndrome 2 with posterior reversible encephalopathy syndrome 88% (7/8) alive and without SCD 13% (1/8) died from disseminated aspergillosis All survivors in school and/or employed Infections 4 with EBV reactivation (2 with PTLD), 1 with CMV enteritis, 6 with HHV-6 reactivation
Foell et al. [52] CD3^+^ and CD19^+^ depletion	PBSC	Myeloablative ATG, fludarabine, thiotepa, and treosulfan GvHD prophylaxis—cyclosporine and MMF	9	89% (8/9) at over a range of 6–42 months (median 26 months)	56% (5/9) grades I–II acute GvHD 11% (1/9) chronic moderate–severe GvHD	100% (9/9)	Grades 1–2 mucositis, diarrhea, limited pain crises with hemiplegia, 1 with neuromuscular spasms with cranial nerve V and VII transient impairment 89% (8/9) alive and without sickle-cell-disease-related symptoms 11% (1/9) died from CMV-pneumonitis Infections 3 with CMV reactivation, 1 with CMV pneumonitis Reactivation of EBV, adenovirus, HHV-6, and BK virus
Bolanos-Meade et al. [6] PTCy	G-BM (3), BM (11)	Non-myeloablative ATG (12 patients), fludarabine, cyclophosphamide, and total body irradiation GvHD prophylaxis—PTCy, FK, sirolimus, and MMF	14 (age range 15–42 years)	100% (14/14) at 7.5–66 months	0% (0/14) acute GvHD 0% (0/14) chronic GvHD	57% (8/14)	50% (7/14) alive and without sickle-cell-related symptoms No new strokes, acute chest syndrome, or priapism Infections 3 with CMV reactivation, 1 with EBV reactivation, and 1 with RSV upper respiratory infection and mycobacterium lung infection
Fitzhugh et al. [7] PTCy	PBSC	Non-myeloablative alemtuzumab, total body irradiation GvHD prophylaxis—PTCy, sirolimus	12 (age range 20–56 years)	92% (11/12)	8% (1/8) acute GvHD 8% (1/8) chronic GvHD	70%	No SCD-related issues and no sinusoidal obstruction syndrome 2 patients with graft rejection developed high-grade myelodysplastic syndrome with fibrosis 1 patient with pulmonary hypertension and heart failure (died) 1 died from infection post-surgery 50% (6/12) alive and without sickle-cell-disease-associated symptoms Infectious 4 with CMV reactivation, 1 with CMV colitis, 1 with disseminated adenovirus, 3 maintained chronic EBV viremia, 1 with EBV-PTLD, 3 were treated for presumed fungal pulmonary nodules, and 15 with bacteremia
De la Fuente et al. [11] PTCy	BM	Non-myeloablative ATG, fludarabine, cyclophosphamide, total body irradiation (all), and thiotepa (15 patients) GvHD prophylaxis—PTCy, MMF, sirolimus	18 (age range 12.1–26 years)	100% (16/16)	13% (2/16) grades III–IV acute GvHD 6% (1/16) limited chronic GvHD	83% (15/18)	1 case of sinusoidal obstruction syndrome 2 with posterior reversible encephalopathy syndrome 1 new infarct (patient who did not engraft) Suspected MMF induced gastritis, ulcer with bleeding, and typhlitis Infections 6 with EBV reactivation (no PTLD), 3 with CMV reactivation, 1 with adenovirus respiratory infection, 1 with BK cystitis, 2 cases of oral HSV infection, 2 with HHV-6 viremia (1 with HHV-6 encephalopathy)

Legend: HLA, human leukocyte antigen; RIC, reduced intensity conditioning; PBSC, peripheral blood stem cell; BM, bone marrow; ATG, anti-thymocyte globulin; G-BM, granulocyte colony-stimulating factor primed bone marrow; MMF, mycophenolate mofetil; PTIS, pre-transplant immune suppression; PTLD, post-transplant lymphoproliferative disorder; PTCy, post-transplant cyclophosphamide; GvHD, graft-versus-host disease; OS, overall survival; EFS, event-free survival; CMV, cytomegalovirus; EBV, Epstein–Barr virus.

**Table 4 jcm-11-04775-t004:** Pros and cons of gene therapy methods for sickle cell disease.

Gene Addition	Mechanism	Pros	Cons
Lentiviral vector gene addition	Lentiviral vector encoding of either a human γ-globin gene or a normal or modified β-globin gene designed for anti-sickling activity Lentiviral vector encoding a short hairpin RNA molecule for posttranscriptional silencing of BCL11A	Stable integration into the host genome for long-term expression No immunogenicity Transduce non-dividing HSCs with high efficiency Can accommodate large transgenes	Semi-random integration leading to potential off-target effects or insertional mutagenesis
**Gene Editing**	**Mechanism**	**Pros**	**Cons**
Nuclease editing (CRISPR/Cas9, ZFN)	NHEJ: HbF induction via disruption of BCL11A erythroid enhancer HbF induction via disruption of BCL11A binding at the gamma globin promoter	Non-integrating Tools are transient High editing efficiency In use in multiple clinical trials	Requires DSB (genotoxicity) Potential off-target editing Induce a p53 response
	HDR: Direct correction of the sickle mutation	Non-integrating Tools are transient High editing efficiency Direct conversion	Requires DSB (genotoxicity) Potential off-target editing Induce a p53 response Requires donor template Lower editing efficiency
Base editing	Direct conversion of the sickle mutation to create Makassar mutation HbF induction by disruption of non-coding regions (BCL11A, gamma globin promoter) or generation of de novo activators (gamma globin promoter)	No DSB Limited insertion/deletions Single or multiplex genome engineering	Potential off-target editing, unwanted bystander editing, or spurious deamination

Legend: CRISPR, clustered regularly interspaced short palindromic repeats; DSB, double-stranded breaks; HSC, hematopoietic stem cell; NHEJ, non-homologous end joining; ZFN, zinc finger nuclease.

## Data Availability

Not applicable.

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
