# Peer review of "Debating the Future of Sickle Cell Disease Curative Therapy: Haploidentical Hematopoietic Stem Cell Transplantation vs. Gene Therapy"

_jcm, 2022, doi:10.3390/jcm11164775_

Round 1

Reviewer 1 Report

In the manuscript entitled " The Future in Sickle Cell Disease Curative Therapy: Haploidentical Hematopoietic Stem Cell Transplantation vs. Gene Therapy," the authors have attempted to summarize the HSCT and gene therapy treatments available to SCD patients to date. However, while the content of this review is of great interest to the scientific community, the manner in which concepts were presented and the overall organization of the review made it very difficult to read. Too often I found myself lost in very long sentences that covered a lot of information. The grammar and occasionally the spelling required serious attention.  I have highlighted problem areas in the manuscript and provided comments and suggestions to help revise the manuscript. Please consider clearly delineating chapters covering HSCT vs gene therapy vs the latest gene editing. As is, there is a lot of repetition and the concepts are unclear. I found it very difficult to read and understand. 

Author Response

Response to Reviewer 1

Line 30: the author referred to “newborn screening, penicillin prophylaxis, and vaccinations” as “disease-modifying therapies”. They do not “modify” the disease in any way. They are only first line of treatment and are not curative, either.

Response: We clarified this sentence to say current disease modifying therapy such as hydroxyurea or chronic transfusions. Newborn screening, penicillin prophylaxis, and vaccinations have improved life expectancy as stated, but disease modifying therapies (clarified) do not eliminate disease manifestations.

Line 31: the author wrote, “ and need to be continued indefinitely”. Suggest changing to: “require lifelong administration”.

Response: We made the change as suggested by the reviewer

Line 33: insert “when” using a human HLA

Response: We made the change as suggested by the reviewer

Line 34: insert “from” matched HLA donors

Response: We made the change as suggested by the reviewer

Line 34: consider starting a new sentence and rephrase “broad use of this option”

Response: We made the change as suggested by the reviewer

Line 40: insert “for patients with SCD”

Response: We made the change as suggested by the reviewer

Line 41: insert “will overcome”

Response: We made the change as suggested by the reviewer

Lines 46-47: I don’t see the point of this statement. I recommend removing it.

Response: We deleted this statement setting up the debate.

Line 48: the author wrote, “at the future for”. Since new gene-editing-based therapies are currently in Phase I/II clinical trials, it would be premature to state that the haplo- HSC is “the future for curative therapies”.

Response: This article is a debate article where the authors take the position that their position (either haploidentical HSCT or gene therapy) will be the future of the curative therapy.

Line 51: replace “donor opportunity” to “donor availability”

Response: We made the change as suggested by the reviewer

Line 52: replace “has” with “have”; replace “of” with “to”

Response: We made the change as suggested by the reviewer

Lines 56-60: very long and convoluted sentence that is very difficult to follow. Break it down into 2 to simplify it.

Response: We modified the sentence for clarity.

Line 60: Obviously “successful” HSCT is a requirement but maybe the author wanted to say “haplo-HSCT”?

Response: We modified the sentence for clarity.

Line 64: replace “remains” with median survival “is “

Response: We made the change as suggested by the reviewer

Lines 70-71: Poor sentence structure. Revise. Ex: it’s not a “viable” but an “alternative” and it is not “across varied age” but “over a wider range of disease phenotypes (severe and milder)”.

Response: We modified the sentence for clarity.

Line 72: add “have shown excellent results in children (include age range)” ; replace “mostly in children using” with “when used in combination with myeloablative conditioning”

Response: We made the change as suggested by the reviewer

Line 74: include reference to “10-15%”; add “the” toxicity

Response: We made the change as suggested by the reviewer

Line 74-75: awkward sentence - Revise.

Response: We made the change as suggested by the reviewer

Lines 86-91: Very long and confusing sentence. Revise.

Response: We deleted this sentence.

Line 93: define what “T-cell replete” is.

Response: We clarified language to define T cell replete.

Line 97: which T-cell subset and reference.

Response: A reference has been added.

Line 102: add reference

Response: References have been added.

Line 122: have significant “and” sometimes overlapping

Response: We made the change as suggested by the reviewer

Lines 124 -125: repetitive – poor sentence structure

Response: We modified the sentence for clarity.

Line 125: Isn’t it obvious that immune recover is

Response: We are unclear what is meant by this comment.

Line 126: replace “was seen” with “was achieved”

Response: We made the change as suggested by the reviewer

Line 131-133: Needs revision. Suggestion: “Infections due to viral reactivation caused by delayed or impaired T-cell mediated immunity are the major complications of TCD and TCR haplo-HSCT”. Is viral reactivation the main and only cause of the infection? Response: We revised this sentence as suggested by the reviewer.

Line 142: This sentence marks the beginning of a new sub-point. All this time you’ve talked about haplo-HSCT and suddenly you’ve start talking about lentivirus. It needs an introduction: for example: why the need to develop a LV gene therapy? It is intended to replace haplo-HSCT? Who would be eligible for LV vs haplo-HSCT – along these lines. Response: Given the debate format of the article, the author is attempting to argue why haplo-HSCT may be more suitable for adults with co-morbidities given initial studies in gene therapy excluded significant comorbidities. We modified this paragraph to make the transition smoother.

Line 145: add reference.

Response: We deleted this portion of the paragraph to make the transition smoother as discussed above.

Line 160: previous paragraph you’ve talked about LV gene therapy and now you start talking again about haplo-HSCT. You need to define a new section otherwise is very confusing: jumping from one subject to another with no clear transition.

Response: To make it clearer that this is a debate style article, we revised the title of the article. We decided to have one author present his argument for haplo-HSCT based on the subheadings (eligibility, conditioning, morbidity, etc) then the second author discusses the same subheadings to argue for gene therapy as the future. Based on the reviewers comments, with which we agree, we revised any discussion of the alternative approach by the author to minimal discussion only; only to argue why one version is potentially better than the other.

Lines 161-165: sentence too long difficult to read. Revise.

Response: We modified the sentence for clarity.

Lines 168-169: the outcome of a transplant is engraftment. GvHD is an unwanted side effect of the procedure. Rather I would state “increase engraftment while decreasing the incidence of GvHD” – sentence needs to be revised.

Response: We made the change as suggested by the reviewer.

Lines 169-170: Revise: 8/10 patients, with a median age of 14 (range: 5-23 years old) underwent haplo-HSCT

Response: We made the change as suggested by the reviewer.

Line 171: add “were reported in 20% of the patients” with only one case of chronic GvHD. Sentence is too long and hard to follow. Consider breaking it down in two.

Response: We made the changes as suggested by the reviewer.

Lines 175-178. Very hard to read too long!

Response: We modified the sentence for clarity.

Lines 194-195: I don’t understand. They received backup cells?

Response: Despite rejecting the transplantation, patients recovered their marrow with autologous hematopoiesis as the Hopkins regimen used a non-myeloablative approach. Back up cells were not required.

Line 195: Revise: “None of the engrafted patients developed GvHD and OS reached 100%”. Was the term “OS” previously defined?

Response: OS was defined previously, now on line 197

Line 199: reference after “graft” and insert “while” reducing donor HLA antibodies Response: We added a reference and modified the sentence as suggested by the reviewer.

Line 203: describe the study better. For example: what were cohorts I-III?

Response: We have modified the sentence to describe cohorts I-III as suggested by the reviewer.

Lines 236-241: seems repetitive. You’ve talked in the above section.

Response: We modified this section to decrease redundancy.

Line 248: insert “an” improved

Response: We made the changes as suggested by the reviewer.

Line 250: add reference there not at the end of the sentence.

Response: We improved the grammar of the sentence.

Lines 269-271: Sentence too long and difficult to read. Revise.

Response: We deleted this paragraph.

Line 274: You’ve start talking about ex vivo genetically modified stem cells without introduction. I’m very confuse.

Response: We deleted this paragraph

Line 281: insert “with” less than half…

Response: We deleted this paragraph

Line 281-282: confusing

Response: We deleted this paragraph.

Line 289: replace with “costs”; replace “with extraordinary” with “resulting in high” Response: We made the changes as suggested by the reviewer.

Line 292: add references.

Response: Reference has been added.

Line 293: “nonelderly lifetime”? what does it mean? Revise.

Response: We modified the sentence for clarity.

Line 302: too long sentence.

Response: We made the change as suggested by the reviewer.

Lines 310-312: revise

Response: We deleted this sentence.

Lines: 316-317 repetitive

Response: We deleted this paragraph.

Line 326: confusing

Response: We modified the sentence for clarity.

Line 331: gene therapy whether is LV or gene editing relies on autologous HSCT – it’s understood that it’s using patient own HSPC. So no need to state 100% donor availability.

Response: We modified this paragraph for clarity and deleted the portion discussing gene therapy.

Line: 336-337: next generation of what?

Response: We deleted this section.

Line 338: what late effects are you referring to? Confusing

Response: We deleted this section.

Line 343: “theoretically available” not true. I believe the field believes that gene-therapy for SCD should be made available to those patients who don’t have a matched or haplo-HLA graft donor or for those who have failed at previous allo-HSCT.

Response: Gene therapy is “theoretically” available to all patients who qualify for transplantation, however per the reviewers point, this section stresses that certain patients are excluded – in particular, patients with HLA-matched siblings are excluded. At this time, there is no evidence to say that haplo-HSCT is preferred and therefore exclusionary to gene therapy  (hence the purpose of this article debating which will be the future). Failed allo-HSCT is an exclusionary criteria for gene therapy.

Lines 345-349: repetition. You’ve talked about this in section 2.

Response: We modified the sentence for clarity.

Line: 350 – rephrase – bad English.

Response: We made the change as suggested by the reviewer.

Line 352: replace “mature” with “becomes available”

Response: We made the change as suggested by the reviewer.

Lines 354-355: I disagree. I believe the statement is incorrect.

Response: We modified the sentence for clarity.

Lines 360-363: Very confusing. Needs revising.

Response: We modified the sentence for clarity.

Line 373: replace “transferred” to “addition”

Response: We made the change as suggested by the reviewer.

Line 375-377: the statement makes no sense. Is there any evidence that there is a proliferative advantage of gene-corrected HSPC over non-corrected one in SCD? If yes, then reduced myeloablative conditioning is something to be considered. Response: We know that hemoglobin normalizes and indicators of hemolysis are improved after gene therapy (Kanter et al, NEJM 2021). This suggests restoration/improvement in red blood cell survival of gene modified cells. Studies are ongoing to measure the red cell survival to answer the question posed in this reviewers comment. We modified the paragraph for clarity.

Line 380-381: I don’t understand – rephrase.

Response: We modified the sentence for clarity.

Line 392: it’s not allogeneic HSCT and gene therapy, it is allogeneic vs autologous HSCT.

Response: We made the change as suggested by the reviewer.

Line 396-400: very long and difficult to follow. I don’t understand what you are trying to say!

Response: We modified the sentence for clarity.

Line 406-421: It is the right place to discuss this but is repetitive. Remove from section 2.

Response: We made the change as suggested by the reviewer.

Line 418: what is “regional gene expression”?

Response: We deleted this for clarity.

Line 426: what is “from the steady-state bone marrow collection”? rephrase.

Response: We deleted this for clarity.

Line 424-429: impossible to read and understand

Response: We modified this paragraph for clarity.

Line 429: what modifications are you referring to?

Response: We modified this paragraph to better describe the modifications.

Line 430-433: make it chronological: start the paragraph with 2021, continue with 2022 then summarize.

Response: We modified this paragraph for clarity.

Line 435: here you state “high levels of modified gene expression” but on line 425 you’ve stated “low modified globin expression” – which one is it? Very confusing and hard to follow.

Response: Given the improvements/modifications made to the trial designs due to initially low globin expression, the current gene therapy products demonstrate high levels of modified gene expression. We modified this sentence for clarity.

Lines 441 and 444: repetition

Response: We made the change as suggested by the reviewer.

Lines 444-445: rephrase. bad sentence structure.

Response: We made the change as suggested by the reviewer.

Line 452: define “F-cells”

Response: We made the change as suggested by the reviewer.

Line 583: Rephrase – bad sentence structure.

Response: We modified for clarity.

Table 4: what is “above + direct conversion”?

Response:

Additional comments:

  1. The subtitles for each section are not always representative of the content of that section. While the sub-title states “haplo-HSCT” the body of the text is described LV gene therapy using autologous HSCT.

Response: We significantly edited the manuscript to delete sections in the haplo-HCST section that pertain to LV gene therapy unless a direct comparison is made.

  1. The review requires major It is very difficult to read and often confusing. Sentence structure is too long with many grammatic and syntactic mistakes.

Response: We greatly appreciate the reviewer’s thorough review of our manuscript. We have made all grammatical and editing suggestions provided by the reviewer and believe it significantly strengthens the manuscript.

Section 2 should have been only about haplo-HSCT and nothing about gene therapy. Overall it was very confusing and difficult to read.

Response: As above, we have made this modification.

Section 3 talks about gene therapy. This section should now be divided into LV- gene therapy and gene-editing-based gene therapy. This would make the review much cleaner and easier to read.

Response: We feel there is not currently enough data to support dividing the argument for gene therapy into LV-gene therapy vs. gene editing based gene therapy. The argument in this manuscript is haplo-HSCT vs. gene therapy rather than arguing for the best the methodology of gene therapy.

Reviewer 2 Report

Overall, the review is comprehensive and will add great value to the readers. Some minor comments to be addressed are included below:

In the abstract, consider revising the word “debate” to discuss.

Clarify the message of the last sentence of the introduction.

Revise the following for grammar/clarity: Currently, children with recurrent stroke (overt and silent), recurrent pain episodes requiring hospitalization not fully managed by medical therapy, recurrent acute chest syndrome episode despite optimal medical therapy, pulmonary hypertension, high blood pressure not managed by medications and those with recurrent priapism, complications associated with long-term poor outcomes, are candidates for curative therapy for HSCT[19-24].

Table 1: clarify “Aforementioned risks”.

“More data for children than adults” Clarify why this is the case. Is it due to a limited ability to enroll adults in the trials?

Table 2: What about mobilized HSPCs as a stem cell source?

Table 2: clarify risk of chip development only “prior” to transplant. It is possible that the transplant procedure induced the development of CHIP “post”-transplant.

Author Response

In the abstract, consider revising the word “debate” to discuss.

Response: We have modified the abstract per the reviewers suggestion.

Clarify the message of the last sentence of the introduction.

Response: Per Reviewer 1's suggestion, we deleted this sentence.

Revise the following for grammar/clarity: Currently, children with recurrent stroke (overt and silent), recurrent pain episodes requiring hospitalization not fully managed by medical therapy, recurrent acute chest syndrome episode despite optimal medical therapy, pulmonary hypertension, high blood pressure not managed by medications and those with recurrent priapism, complications associated with long-term poor outcomes, are candidates for curative therapy for HSCT[19-24].

Response: We modified the sentence for clarity.

Table 1: clarify “Aforementioned risks”.

Response: We clarified as suggested by the reviewer.

“More data for children than adults” Clarify why this is the case. Is it due to a limited ability to enroll adults in the trials?

Response: There is one study reporting this methodology and it includes only children. We clarified language in the table.

Table 2: What about mobilized HSPCs as a stem cell source?

Response: We clarified as suggested by the reviewer.

Table 2: clarify risk of chip development only “prior” to transplant. It is possible that the transplant procedure induced the development of CHIP “post”-transplant.

Response: We clarified this point in the manuscript (paragraph lines 550-566)

Round 2

Reviewer 1 Report

Dear Authors,

Thank you for addressing my concerns and for revising the manuscript.  I found it to be significantly improved from the previous version. I only suggest revising the sentence in lines 152 and 449.

Author Response

We thank the reviewer for his/her thorough and thoughtful corrections to the manuscript. We believe it has strengthened the manuscript significantly. We have made the modifications to lines 152 and 449 as suggested.